# Sodium hypochlorite inactivation of human CJD prions

**Bradley R. Groveman**[☯]*, **Brent Race**[iD][☯]*, **Andrew G. Hughson, Cathryn L. Haigh**

The Laboratory of Neurological Infections and Immunity, Rocky Mountain Laboratories, National Institute of Allergy and Infectious Diseases, National Institutes of Health, Hamilton, Montana, United States of America

☯ These authors contributed equally to this work.
* raceb@niaid.nih.gov (BR); bradley.groveman@nih.gov (BRG)

**Data Availability Statement:** All relevant data are within the manuscript and its Supporting Information files.

**Funding:** This research is funded by the Division of Intramural Research, NIAID/NIH.

## Abstract

Prion diseases are transmissible, fatal neurologic diseases of mammals caused by the accumulation of mis-folded, disease associated prion protein (PrPd). Creutzfeldt-Jakob Disease (CJD) is the most common human prion disease and can occur by sporadic onset (sCJD) (~85% of CJD cases), genetic mutations in the prion protein gene (10–15%) or iatrogenic transmission (rare). PrPd is difficult to inactivate and many methods to reduce prion infectivity are dangerous, caustic, expensive, or impractical. Identifying viable and safe methods for decontamination of CJD exposed materials is critically important for medical facilities and research institutions. Previous research has shown that concentrated sodium hypochlorite (bleach) was effective at inactivation of CJD prions derived from brains of mice or guinea pigs. Unfortunately, human prions adapted to rodents may mis-fold differently than in humans, and the rodent adapted prions may not have the same resistance or susceptibility to inactivation present in bona fide CJD prions. To confirm that bleach was efficacious against human sourced CJD prions, we exposed different subtypes of sCJD-infected human brain homogenates to different concentrations of bleach for increasing exposure times. Initial and residual prion seeding activity following inactivation were measured using Real-Time Quaking Induced Conversion. In addition, we tested how passage of human sCJD into either transgenic mice that expressed human prion protein, or transmission of CJD to human cerebral organoids (CO), two common laboratory practices, may affect CJD prions' susceptibility to bleach inactivation. Our results show that bleach is effective against human sourced sCJD prions, and both treatment time and concentration of bleach were important factors for CJD inactivation. CJD derived from human brains, transgenic mouse brains or CO were all susceptible to inactivation with as low as a 10 percent bleach solution with a 30-minute exposure time or a 50 percent bleach solution with as little as a 1-minute exposure time.

## Introduction

Inactivation of infectious prions (prions) has been a challenge and concern for research laboratories, medical facilities, and meat processing plants for many years. Many disinfectants

**Competing interests:** The authors have declared that no competing interests exist.

marketed for destruction of bacterial and viral pathogens are not effective in eliminating prion infectivity. Several chemical disinfectants have been identified that do inactivate prions [1] but these chemicals are often dangerous to handle or very damaging to equipment. The most effective and frequently used methods for decontaminating prions are exposure to either concentrated sodium hydroxide or concentrated sodium hypochlorite. However, these are highly caustic and present biosafety hazards of their own.

We recently tested the ability of two phenolic disinfectants, Environ LpH (a CDC approved prion decontaminant [1]) and Wex-cide 128, to inactivate prions derived from humans, rodents and deer. Our results showed that both Environ LpH and Wex-cide 128 eliminated several logs of prion infectivity from brain homogenates infected with either chronic wasting disease (CWD) prions or rodent-adapted scrapie but performed poorly against human Creutzfeldt-Jakob Disease (CJD) prions where only slightly more than one log of inactivation was achieved [2].

The inconsistency of inactivation between prions from different strains or species was not a unique observation. Previous studies have also reported that different prions can differ in resistance to inactivation [3–9]. Unfortunately, human trophic CJD appears to be among the most difficult to inactivate and was 10,000–100,000 times more resistant to inactivation by either acidic SDS or Wex-cide 128 compared to rodent scrapie [2, 3, 6]. This suggests that specific decontamination methods may need to be validated for efficacy against the prion strain in question.

Sporadic CJD presents as a variety of subtypes. These subtypes are primarily defined by a combination of shifts in the mobility of their PrPd protease resistant core (eg: Type 1 or Type 2), as observed by electrophoresis and western blotting [10], and a polymorphism at codon 129 of the prion protein gene, encoding for either methionine (M) or valine (V) [11, 12]. Different subtypes present with different symptomology and disease course [13], and as such may present different resistances to inactivation. It is therefore important to identify decontamination methods effective against the different sCJD subtypes.

Historically, treatment using concentrated sodium hypochlorite (bleach) solutions were reported to eliminate 3–4 logs of CJD infectivity [14–16]. At the time these initial bioassay studies were performed, genetically engineered mice expressing human prion protein that can be used to accurately model human prion infections, were not available. As an approximate model, sCJD that had been passaged and adapted to guinea pigs [14, 15] or mice [14, 16] was used for inactivation experiments. Unfortunately, both guinea pigs and mice have very different prion protein amino acid sequences compared to humans, and adaptation of a prion agent from one species to another may not accurately confer the original biochemical properties or susceptibility to inactivation associated with prions from the initial host species [3, 17]. Since the initial studies were done with rodent adapted CJD and not with bona fide human CJD prions, it is prudent to confirm that human derived CJD can also be inactivated using similar bleach treatments. Fortunately, recent studies by Belondrade et al. have reported bleach conditions to inactivate human derived vCJD [7, 8] and Belondrade et al. and Mori et al. have shown that 1N NaOH is an effective method to inactivate both vCJD and sCJD, respectively [8, 18]. However, data that directly tests bleach against human derived sCJD is still unavailable.

In the current study, we determined the necessary times and concentrations of bleach required to inactivate sCJD prions derived directly from human brains infected with the four most prevalent subtypes of sCJD (MM1, MV1, MV2 and VV2) [19]. Although MM1 and MV1 behave similarly in animal transmission studies and therefore have been described as a unified subtype (M1) [20], for clarity in this manuscript we use the Parchi et al. classification system of molecular subtypes [10] to distinguish between the samples tested. In addition to human brain

tissues, we evaluated the effectiveness of bleach against MV1 and MV2 sCJD prions that had been passaged once or twice in transgenic mice that express human prion protein, as well as against prions from cultured human cerebral organoids infected with MV2 sCJD. Prion inactivation was measured by monitoring the loss of self-propagation, or "seeding", capacity of both untreated and bleach treated tissues using ultrasensitive Real-Time Quaking Induced Conversion (RT-QuIC). While the gold-standard for prion inactivation has historically been animal bioassay, RT-QuIC has been shown to be at least as sensitive as end-point bioassays [21]. Both RT-QuIC and the alternative prion seed amplification assay Protein Misfolding Cyclic Amplification (PMCA), have recently been utilized as correlates for bona fide prion infectivity and have become extremely useful assays for decontamination applications [8, 18, 21–25]. This study not only identifies appropriate conditions for inactivating sCJD prions with bleach, which will allow research facilities and medical, surgical, and clinical laboratories that work with sCJD infected patients and tissues to design appropriate decontamination protocols, but also provides a model for testing future decontaminants against different strains and subtypes of prions.

## Materials and methods

### Ethics statement

The human brain and brain organoid homogenates used in this study were from samples and cells de-identified before being provided to the researchers at the NIH. Thus, the NIH Office of Human Subjects Research Protections (OHSRP) has determined these samples to be exempt from IRB review. Mouse passaged CJD was obtained from historical stock brain homogenates from a previous study [26] approved by the Institutional Animal Care and Use Committee of the Rocky Mountain Laboratories (RML) protocols 2019–007-E and 2022–006-E.

### Decontamination of CJD homogenates using bleach

Pure bright brand bleach containing 6% sodium hypochlorite was used in all bleach experiments. We considered undiluted bleach to be 100% bleach and the bleach dilutions tested here are based on this value, not the percentage of sodium hypochlorite. We screened CJD brain homogenates derived directly from four different donors with human CJD (MM1, MV1, MV2, and VV2), brain homogenates from transgenic mice (tg66) homozygous for the M129 human prion protein inoculated with the same MV1 and MV2 afflicted human brain homogenates as well from a second passage of this material, and MV2 CJD derived from a human cerebral organoid infected with the same MV2 brain homogenate as above [26, 27].

Ten microliters of a 10% tissue homogenate was mixed with 90 μl of several different concentrations of diluted bleach for a final concentration of 1% tissue homogenate and 1%, 10%, 20%, or 50% bleach for either 1, 5 or 30 minutes. The final chlorine concentrations in parts per million (ppm) were 450, 4,500, 9,000 and 22,500 respectively. Immediately following the treatment, samples were diluted 100-fold into RT-QuIC diluent (0.1% SDS, 1x N-2 supplement, 1x PBS). Untreated controls were diluted directly into RT-QuIC diluent. The 100-fold dilution was necessary to reduce any potential residual anti-prion activity of the bleach and decrease the bleach concentration to levels previously shown not to inhibit the RT-QuIC assay [25]. Diluted samples were loaded directly into the RT-QuIC assay following dilution.

### RT-QuIC assay

RT-QuIC reactions were performed as previously described [27]. One microliter of the indicated tissue dilution or diluted bleach treated sample from above were loaded into a black

384-well clear bottom plate (Nunc) containing 49 μl of RT-QuIC reaction mix. Seeded reaction mix contained a final concentrations of 0.002% SDS, 10 mM phosphate buffer (pH 7.4), 300 mM NaCl, 0.1 mg/ml recombinant hamster 90–231 (Ha rPrP) (accession no. KO2234) [21, 28], 10 μM thioflavin T (ThT), and 1 mM ethylenediaminetetraacetic acid tetrasodium salt (EDTA). Four to 8 replicate reactions were tested per sample. The plate was then sealed with a plate sealer film (Nunc) and incubated at 50˚C in a BMG FLUOstar Omega plate reader with a repeating protocol of 1 min shaking (700 rpm double orbital) and 1 min rest for the 30 hr reaction time. ThT fluorescence measurements (450 +/-10 nm excitation and 480 +/-10 nm emission; plate bottom read) were taken every 45 min. Individual wells were considered positive if their baseline subtracted maximum ThT fluorescence level reached greater than 10% of the baseline subtracted maximum ThT fluorescence value on the entire reaction plate within the 30 hr reaction time.

### Estimation of prion seeding activity

Seeding dose 50 ($SD_{50}$) titers were calculated for each untreated brain or organoid homogenate by end-point dilution analysis using the Spearman-Kärber formula [21]. Prion infectivity titers for the brain homogenate experiments are reported as the $\log_{10}SD_{50}$ per mg of diluted brain.

For estimating the reduction in seeding activity due to bleach treatments the end-point dilution values for each CJD-infected homogenate were used to generate standard curves (S1 Fig). The log-based fold dilution (i.e.: 3, 4, 5, etc.) was plotted (x-axis) versus percent positive wells (y-axis). For each untreated CJD-infected homogenate, a simple linear regression line was drawn from the last dilution with 100% positive wells through the first dilution with 0% positive wells. The percent positive well values for the treated samples were then entered into the linear equation generated by the regression to calculate the corresponding log dilution of the untreated sample. This value represents the theoretical maximum log reduction resulting from the treatment (Tables 1 and 2). However, since the treated samples were tested at a $10^{-4}$ dilution it could not be determined whether, if tested at a lesser dilution, some reaction wells would show positive seeding activity. Therefore, we subtracted 4 logs from the theoretical maximum ($10^0$ dilution or 100% tissue) to calculate the minimum observed log reduction (MinLR) resulting from each treatment (Figs 1C and 2C and Tables 1 and 2).

## Results

### Efficacy of bleach against four subtypes of human derived sCJD

The ability of disease associated prions (PrPd) to induce misfolding of, or "seed", native prion protein (PrPc) is the basis for seed amplification assays such as the Real-Time Quaking Induced Conversion assay (RT-QuIC). The ultrasensitive RT-QuIC can detect femto- or atto-

**Table 1.  Titers and log reductions in treated human sCJD brain homogenates.**

|      | $\log SD_{50}$/mg | MaxLR* | MinLR$ | RR# |
|------|------|------|------|------|
| MM1 | 5.5 | 6.9 | 2.9 | 0.5 |
| MV1 | 6.75 | 7.8 | 3.8 | 0.6 |
| MV2 | 7.5 | 8 | 4 | 0.5 |
| VV2 | 6.25 | 7.2 | 3.2 | 0.5 |

*maximum log reduction

$minimum log reduction

#log reduction ratio (MinLR:logSD50/mg of the untreated brain homogenate)

**Table 2. Titers and log reductions in treated tissues from passaged sCJD material.**

|            | $logSD_{50}$/mg | MaxLR* | MinLR$ | RR# |
|------------|-----------------|--------|--------|-----|
| MV1 (P1)   | 6.5             | 7.9    | 3.9    | 0.6 |
| MV1 (P2)   | 7               | 7.7    | 3.7    | 0.5 |
| MV2 (P1)   | 7.75            | 8.8    | 4.8    | 0.6 |
| MV2 (P2)   | 7.25            | 8.2    | 4.2    | 0.6 |
| MV2 (Org)  | 5.13            | 6.7    | 2.7    | 0.5 |

*maximum log reduction

$minimum log reduction

#log reduction ratio (MinLR:logSD50/mg of the untreated brain homogenate)

gram amounts of PrPd in a prion contaminated sample [21, 29]. In biological samples, seeding activity strongly correlates with infectivity [21], making it a strong readout for testing decontamination procedures. To determine the efficacy of bleach as a decontaminant for sCJD prions we used the RT-QuIC to monitor the reduction of seeding activity following treatment with various concentrations of bleach and increasing contact times. CJD prions from prion-infected human brain homogenates (BH; 1% w/v), were exposed to 1, 10, 20 or 50% household bleach for 1, 5, or 30 min.

In general, increasing either the concentration or contact time of the bleach resulted in greater reductions in seeding activity (Fig 1). No striking differences were observed in susceptibility to bleach decontamination between the 4 sCJD subtypes tested here. In all cases 1% bleach was unable, even at the longest contact time, to remove all seeding activity, and in some cases, seeding was still observed in all replicate wells (Fig 1A). Thirty-minute contact times with 10% bleach, 5 min or greater contact times with 20% bleach, and 1 min or greater contact times with 50% bleach each completely removed all detectable seeding activity from all 4 subtypes (Fig 1A). Although not all of the conditions were able to remove all seeding activity, wells that maintained seeding activity after treatments displayed reductions in the aggregation kinetics of the active wells compared to the untreated controls (Fig 1B). This reduction was apparent even after a 1 min treatment with 1% bleach. The average log reduction across the subtypes in conditions where bleach treatment eliminated all seeding activity were 3.5±0.5–7.5±0.5 logs (Table 1). Although the calculated log reductions differ between subtypes (Figs 1C and S1), when the minimum log reduction (MinLR) for each subtype is compared with the titer of the inocula ($logSD_{50}$/mg; Table 1) the reduction ratios (MinLR with no positive wells:$logSD_{50/mg}$) appear similar: 0.5, 0.6, 0.5, and 0.5 for MM1, MV1, MV2, and VV2, respectively. Overall, this demonstrates that bleach is an effective decontaminant for sCJD prions from human brain tissue.

## Efficacy of bleach against passaged sCJD prions

To investigate whether the decontamination efficiency of bleach may change when human sCJD prions are passaged in either tg66 mice expressing human PrP or human cerebral organoids, we performed similar decontamination experiments using prion infected tissues from these sources. Tissues included brain homogenates from tg66 mice that were inoculated with the same MV1 and MV2 brain homogenates as tested above (P1), a second passage (P2) of brain material from the P1 mice, and MV2 human brain homogenate passaged through human cerebral organoids (CO) (organoid tissue generated in [27]; mouse tissue generated in [26]).

Decontamination of the passaged sCJD prions was similar to that of the original human brain tissue (Fig 2). Again, 1% bleach was ineffective at removing all seeding activity, but 30 min contact times with 10% or 20% bleach and 1 min or greater contact times with 50% bleach

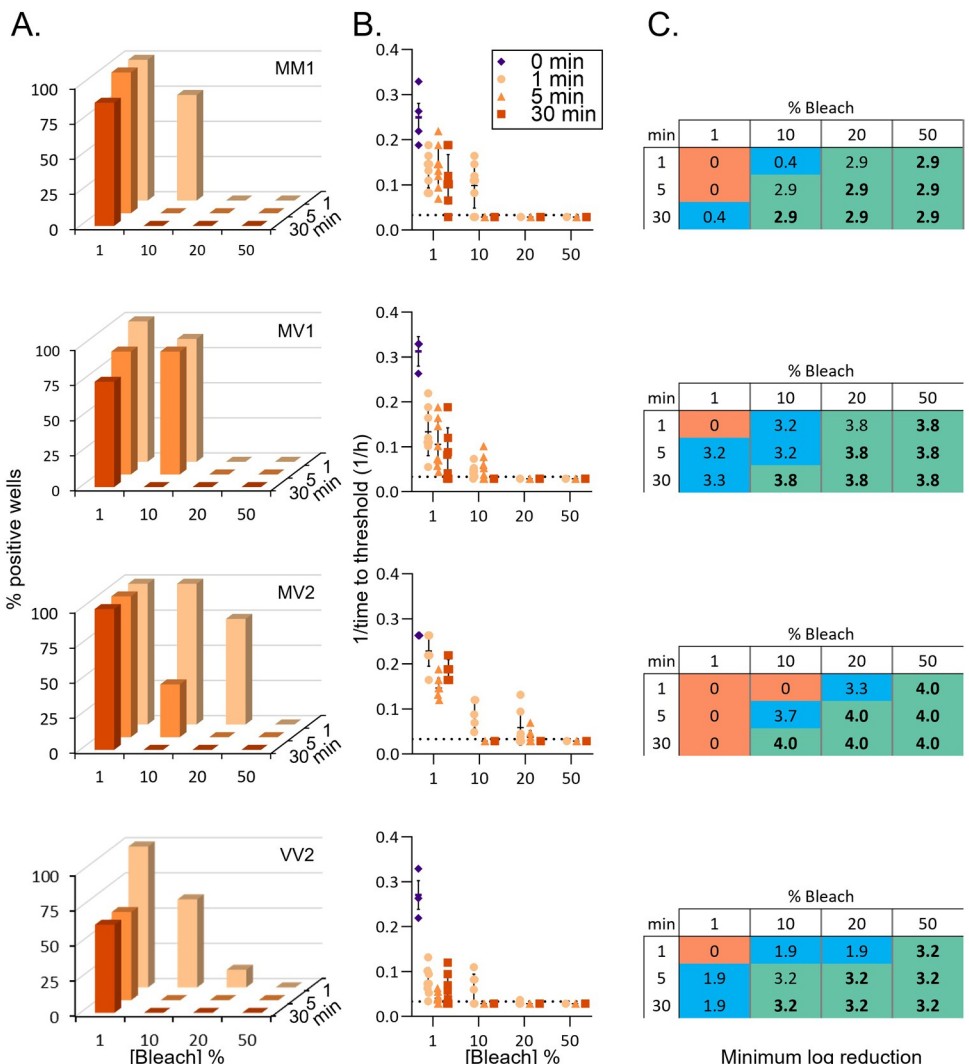

**Fig 1. Inactivation of prion seeding activity in sCJD infected human brain homogenates.** A. Percent positive reaction wells following increasing treatment times with increasing concentrations of bleach. B. Inverse time to threshold kinetics for each individual reaction following increasing treatment times and concentrations. The 0 min condition refers to testing of the same dilution of untreated brain homogenate. C. Estimated log reductions in seeding activity displayed as minimum log reduction (see methods). Orange, blue, and green highlighting indicate no detectable loss, partial loss, or complete loss of seeding activity, respectively. Bold numbers indicate treatment conditions that resulted in total loss of seeding activity across all sCJD subtypes. Row 1, MM1; Row 2, MV1; Row 3, MV2; Row 4, VV2.

each completely removed all detectable seeding activity from all 4 subtypes (Fig 2A). Shorter contact times were more variable with 20% bleach, but overall, still showed a strong reduction in seeding activity (Fig 2A and 2B). Similar results were obtained for both P1 and P2 indicating the additional subpassaging of the material did not greatly influence its susceptibility to bleach. Estimated log reductions in conditions where bleach treatment eliminated all seeding activity were also similar between human brain homogenate, P1, and P2, with log reductions of 3.8–7.8, 3.9–7.9 and 3.7–7.7 for MV1 and 4.0–8.0, 4.8–8.8 and 4.2–8.2 for MV2, respectively (Figs 1C and 2C and Tables 1 and 2). This resulted in log reduction ratios of 0.6, 0.6, and 0.5 for MV1 human brain homogenate, P1, and P2 and 0.5, 0.6, and 0.6 for MV2 human brain homogenate, P1, and P2 (Figs 2C and S1 and Tables 1 and 2). The human CO passaged MV2

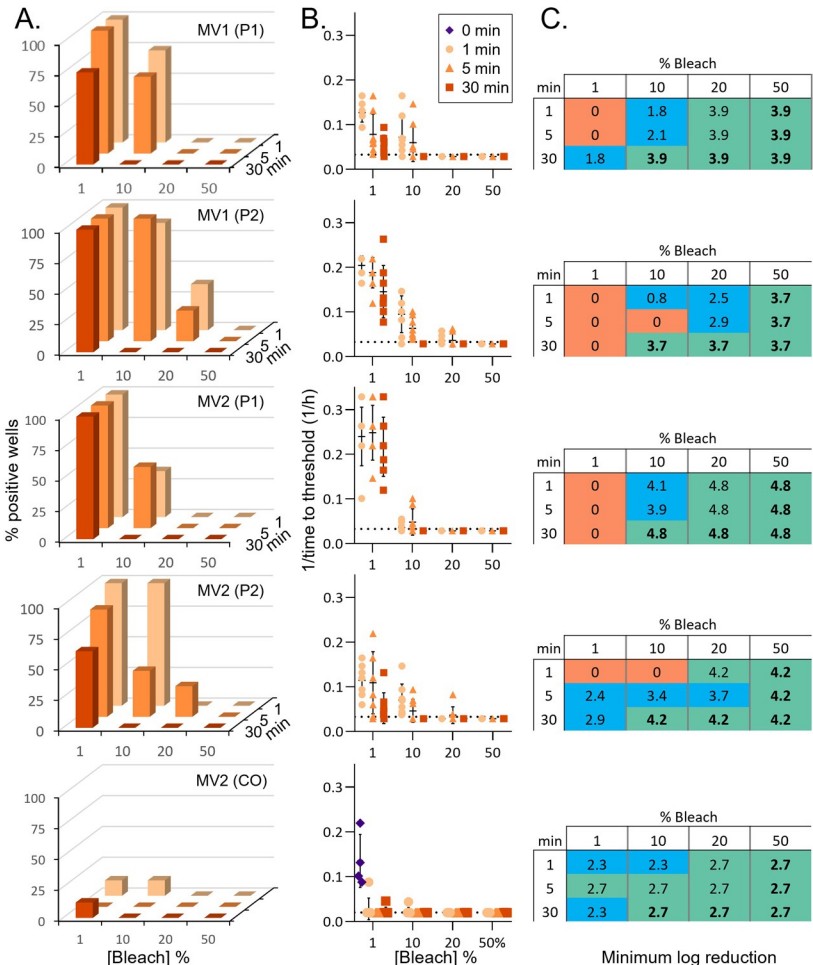

**Fig 2. Inactivation of prion seeding activity in tissues from passaged sCJD material.** A. Percent positive reaction wells following increasing treatment times with increasing concentrations of bleach. P1 indicates brain tissue from a mouse inoculated with human sCJD brain material. P2 indicates brain tissue from a mouse inoculated with P1 material. CO indicates a human cerebral organoid inoculated with human sCJD brain homogenate. B. Inverse time to threshold kinetics for each individual reaction following increasing treatment times and concentrations. The 0 min condition refers to testing of the same dilution of untreated brain homogenate. This is only displayed for MV2 (CO) as the equivalent dilutions were not tested on the reaction plates with the P1 and P2 treatments. C. Estimated log reductions in seeding activity displayed as minimum log reduction (see methods). Orange, blue, and green highlighting indicate no detectable loss, partial loss, or complete loss of seeding activity, respectively. Bold numbers indicate treatment conditions that resulted in total loss of seeding activity across all passaged sCJD subtypes. Row 1, MV1(P1); Row 2, MV1(P2); Row 3, MV2(P1); Row 4, MV2(P1); Row 5, MV2(CO).

material showed a lower log reduction of 2.7–6.7, likely due to a lower starting titer (logSD$_{50}$/mg of 5.1) but had a similar log reduction ratio of 0.5 (Fig 2 and Table 2). Together, these data indicate that passaging human sCJD prions though mice or organoids that express human PrP does not significantly change the susceptibility of the prions to decontamination by bleach.

## Discussion

The finding that sCJD prions are less susceptible to different types of decontamination, such as phenolic compounds [2] or acidic sodium dodecyl sulfate [6], than other prions raises concerns about the efficacy of these, and other decontamination methods extrapolated only from testing on non-human prions or human prions passaged through other models. Here we

demonstrate that bleach can effectively inactivate human brain derived sCJD prions independent of subtype or passage through other models. Concentration of bleach and contact time are inversely related such that one could be raised and the other reduced while maintaining similar efficacy. This could allow for protocols to be modified based on the level of infectivity present, or the type of material or equipment being decontaminated.

While prion seeding activity is not a direct measure of infectivity, a strong link has been found between the two, with higher seeding titers correlating to higher infectivity, across multiple strains of prions and across different species [8, 21, 23, 30–32]. Additionally, seeding assays have been shown to be at least as sensitive, and in some cased up to 4-logs more sensitive, than bioassay [32] and a more reliable marker for infectivity than levels or PK-resistant PrP [30, 31]. Given the close relationship between loss of seeding activity and loss of infectivity, and with bioassays for sCJD taking 180 to 450 days depending on the strain [26], seeding activity has become a widely used surrogate for decontamination studies [7, 8, 24, 25, 33, 34].

In this study, bleach treatments were performed on 1% brain homogenates. However, a 100-fold dilution was required for testing by RT-QuIC and therefore brain homogenates were tested at 0.01%. Because of this 4-log dilution from the starting brain tissue we cannot exclude the possibility that some seeding activity might persist at lesser dilutions. However, even accounting for this possibility, we still observed minimum reductions of ~3–4 logs of prion seeding activity and up to ~7–8 log theoretical maximum reductions among the different human sCJD brain homogenates tested.

Our data are reassuring in the context of the existing prion decontamination guidelines in the 6th edition of Biosafety in Microbiological and Biomedical Laboratories [1]. These guidelines suggest 20,000 ppm bleach (equivalent to 44.4% in the current study) for 1 hour. Based on our findings, researchers adopting these recommendations would inactivate sCJD prions. However, it may be worth revisiting aspects of these guidelines. As stated above, the inverse relationship between bleach contact time and concentration may allow for more delicate items to be decontaminated at a lower concentration resulting in less corrosion. Furthermore, none of the sCJD or passaged sCJD samples required longer than 30 minutes for seeding activity to reach undetectable levels even at the 10% dilution. An initial study defining the length of bleach treatment only treated samples for 15 or 60 minutes and used animal adapted CJD, so it could have appeared to require longer to decontaminate than is genuinely needed for sCJD tissues [15].

This and other studies stress the importance of testing decontamination methods for the specific type of prion and system being used. For example, we recently demonstrated that while bleach is effective at inactivating CWD prions on stainless steel surfaces, the same treatment is ineffective on intact pieces of infected tissue. Similarly, wet tissue might be more susceptible to decontamination than prions dried to a surface [25]. In this context, the use of a 1 minute, 1:10 dilution of bleach to treat an un-broken skin exposure to human prion autopsy tissue, although secondary to washing the site as described in [1], might be reconsidered as these conditions were not sufficient to remove sCJD prions in any of the materials tested here.

Decontamination methods are becoming easier to test with the use of seed amplification assays such as the RT-QuIC, reducing the need for long and expensive animal bioassays. Furthermore, recent methods have been developed for testing large surfaces for prion contamination and the efficiency of prion decontamination methods [33, 35, 36], as well as for using surrogates for stainless steel medical or farm instruments, such as wires or beads, to test decontamination protocols [25, 37]. These tools provide important platforms to determine the efficacy of decontamination methods quickly and sensitively. While it is still unclear why human prions are more resistant to decontamination than other animal derived prions, this study

demonstrates that household bleach can effectively inactivate human CJD prions given the proper concentrations and contact times.

## Supporting information

**S1 Fig. Example log reduction calculations.** An example calculation is shown where the simple linear regression is drawn from the percent positive wells at each 10-fold dilution of an untreated brain homogenate. The equation is displayed above the curve. The example in gray is for a sample that following treatment displays 7 out of 8 positive wells (87.5% positive). Inserted into the equation this gives the equivalent of a 4.4 log reduction (MaxLR). However, since the sample was tested at a 4-log dilution, the possibility cannot be excluded that seeding activity might still be present at a lesser, untested dilution. Therefore, the minimum log reduction (MinLR) is calculated as the MaxLR minus the 4 logs that could not be tested. The actual log reduction for each treatment may fall at or between the MaxLR and MinLR. The log reduction ration (RR) is calculated as (MinLR with no positive wells) / (logSD$_{50}$/mg brain tissue) to account for differences in starting seeding titer of the brain material (Tables 1 and 2).
(TIF)

## Acknowledgments

We thank Dr. Suzette A. Priola for critical review of the manuscript.

## Author Contributions

**Conceptualization:** Bradley R. Groveman, Brent Race, Cathryn L. Haigh.

**Data curation:** Bradley R. Groveman.

**Formal analysis:** Bradley R. Groveman, Brent Race, Cathryn L. Haigh.

**Funding acquisition:** Cathryn L. Haigh.

**Investigation:** Bradley R. Groveman, Brent Race.

**Methodology:** Bradley R. Groveman.

**Project administration:** Brent Race.

**Resources:** Andrew G. Hughson.

**Writing – original draft:** Bradley R. Groveman, Brent Race.

**Writing – review & editing:** Bradley R. Groveman, Brent Race, Cathryn L. Haigh.

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
