## [Decision Letter · Decision Letter 0]

27 Aug 2024

PONE-D-24-29744Sodium hypochlorite inactivation of human CJD prionsPLOS ONE

Dear Dr. Race,

Thank you for submitting your manuscript to PLOS ONE. After careful consideration, we feel that it has merit but does not fully meet PLOS ONE’s publication criteria as it currently stands. Therefore, we invite you to submit a revised version of the manuscript that addresses the points raised during the review process.

We look forward to receiving your revised manuscript.

Kind regards,

Gianluigi Zanusso

Academic Editor

PLOS ONE

Journal requirements: 1. When submitting your revision, we need you to address these additional requirements. Please ensure that your manuscript meets PLOS ONE's style requirements, including those for file naming. The PLOS ONE style templates can be found at https://journals.plos.org/plosone/s/file?id=wjVg/PLOSOne_formatting_sample_main_body.pdf and https://journals.plos.org/plosone/s/file?id=ba62/PLOSOne_formatting_sample_title_authors_affiliations.pdf. 2. We note that the grant information you provided in the ‘Funding Information’ and ‘Financial Disclosure’ sections do not match.  When you resubmit, please ensure that you provide the correct grant numbers for the awards you received for your study in the ‘Funding Information’ section. 3. Thank you for stating the following financial disclosure:  [This research is funded by the Division of Intramural Research, NIAID/NIH.  ].  Please state what role the funders took in the study.  If the funders had no role, please state: ""The funders had no role in study design, data collection and analysis, decision to publish, or preparation of the manuscript."" If this statement is not correct you must amend it as needed. Please include this amended Role of Funder statement in your cover letter; we will change the online submission form on your behalf. 4. Your ethics statement should only appear in the Methods section of your manuscript. If your ethics statement is written in any section besides the Methods, please delete it from any other section. 5. Please include your tables as part of your main manuscript and remove the individual files. Please note that supplementary tables (should remain/ be uploaded) as separate ""supporting information"" files". 6. Please include captions for your Supporting Information files at the end of your manuscript, and update any in-text citations to match accordingly. Please see our Supporting Information guidelines for more information: http://journals.plos.org/plosone/s/supporting-information. 

Reviewers' comments:

Reviewer's Responses to Questions

**Comments to the Author**

1. Is the manuscript technically sound, and do the data support the conclusions?

Reviewer #1: Yes

Reviewer #2: Yes

2. Has the statistical analysis been performed appropriately and rigorously? 

Reviewer #1: N/A

Reviewer #2: Yes

3. Have the authors made all data underlying the findings in their manuscript fully available?

Reviewer #1: Yes

Reviewer #2: Yes

4. Is the manuscript presented in an intelligible fashion and written in standard English?

Reviewer #1: Yes

Reviewer #2: Yes

5. Review Comments to the Author

Reviewer #1: The manuscript by Groveman, Race and colleagues investigates bleach efficacy inactivating human CJD prions. The main finding is that bleach is effective in a dose and time-dependent manner. The experimental design is coherent with the aims, including the use of the RT-QuIC assay as surrogate measure of prion infectivity, the results are clear, and overall, the manuscript is well-written. The topic is timely and provide relevant information about treatment of sources contaminated with human prions especially for the laboratory setting.

I have only few concerns.

There is inconsistency in the exchangeable use of the terms “CJD subtypes” and “CJD prions”. MM1 and MV1 are a unique histo-molecular entity (= MM1/MV1 subtype), and transmission studies confirmed their relationship with the same (M1) prion strain. MV2 and VV2 are different CJD subtypes but behave as a unique prion strain (V2) (see https://doi.org/10.1073/pnas.1004688107). Therefore, in the present manuscript the authors evaluated three CJD subtypes (MM1/MV1, MV2 and VV2) and two prion strains (M1 and V2).

In the Method section, it is unclear:

1-how many samples/cases did the authors used for each experiment set?

2-What brain region(s) did the authors sampled for brain homogenates (i.e., CJD subtypes often show a different PrPd distribution)

3-Did the authors analyzed the PrPd concentration in brain homogenates before beach treatment?

Please, check Jacob -> Jakob.

Reviewer #2: In this manuscript, Groveman and colleagues describe the results of an interesting study on the sensitivity of sporadic Creutzfeldt-Jakob disease (sCJD) human prions to bleach inactivation, one of the few effective chemicals for prion decontamination recommended by national and international guidelines.

Authors treated with sodium hypochlorite (9 treatment conditions, 3 times x 3 concentrations) a panel of tissue suspensions derived from sCJD patients and from sCJD-infected humanized mice and organoids. To quantify residual prion infectivity, they measured the amyloid seeding capacity of each sample by the Real-time quaking induced conversion assay (RT-QuIC).

Authors convincingly show that the seeding properties of human and derived prion strains may be fully cleared after the application of bleach, even at conditions that are less stringent than those reported in official guidelines (20.000 ppm per 1 hour; e.g. https://iris.who.int/bitstream/handle/10665/66707/WHO_CDS_CSR_APH_2000.3.pdf?sequence=1&isAllowed=y; Meechan PJ, Potts J. Biosafety in microbiological and biomedical laboratories. 6th ed. Washington, DC: Department of Health and Human Services; 2020). These results remark the usefulness of bleach to improve safety in laboratory, hospital, or industrial settings that have to deal with prion infectivity and provide a novel set of information that, if confirmed and enlarged by further studies, may help, in the future, to develop dedicated decontamination protocols.

The quality and the appropriatedness of the manuscript are beyound any doubt, yet it has to be underlined that its immediate practical consequences, in a delicate field like that of human prion diseases, are limited by the absence of experimental data that relates the RT-QuIC seeding capacity and the “true” infectious titre of sCJD prions for humans.

Also, as bleach treatment may alter RT-QuIC prion seeding capacity in a way that is not directly proportional to its effect on prion infectivity, it would have been useful to compare residual seeding capacity and prion infectivity in at least two dilutions of the same samples after treatment with bleach, following, for example, the approach already used by these authors in previous publications (Hughson AG, Race B, Kraus A, Sangare LR, Robins L, Groveman BR, et al. Inactivation of Prions and Amyloid Seeds with Hypochlorous Acid. PLoS Pathog. 2016;12(9):e1005914).

Some comments on this topic are welcome.

Another issue is linked to the experimental design. In spite of the high sensitivity of RT-QuIC, the experimental setting only allowed to demonstrate a maximum removal of 4 logs of seeding activity for human brain prions and 4.8 logs for mouse-passaged human prions.

All that said, I recommend the publication of this paper with two main suggestions to authors: the first, to remove the theoretical values (in parentheses) from tables in figures 1 and 2 and, the second, to correctly report Alfons Jakob (not Jacob) surname. Only one minor note: in line 151, please specificy the number of supplemental figure.

6. PLOS authors have the option to publish the peer review history of their article (what does this mean?). If published, this will include your full peer review and any attached files.

Reviewer #1: No

Reviewer #2: **Yes: **Franco Cardone

---

## [Author Response · Author response to Decision Letter 0]

2 Oct 2024

We would like to thank the editor for the opportunity to submit our revised manuscript as well as the reviewers for taking the time to provide constructive comments for us to improve it. We have addressed each comment below point by point. You can find our responses below the corresponding comment. A word doc has also been uploaded with our responses shown in red to facilitate reading.

Editor comments/Journal requirements (see email from editor to us for context of queries):

1. We have corrected the style and file naming to fit PLOS ONE.

2. No grants were received, the financial sections now match.

3. The funders had no role in the study, so please insert “The funders had no role in study design, data collection and analysis, decision to publish, or preparation of the manuscript.” in the online submission form. Thank you.

4. The extra ethics statement has been removed.

5. Tables 1&2 are now in the manuscript file. 

6. Done

Reviewer #1: The manuscript by Groveman, Race and colleagues investigates bleach efficacy inactivating human CJD prions. The main finding is that bleach is effective in a dose and time-dependent manner. The experimental design is coherent with the aims, including the use of the RT-QuIC assay as surrogate measure of prion infectivity, the results are clear, and overall, the manuscript is well-written. The topic is timely and provide relevant information about treatment of sources contaminated with human prions especially for the laboratory setting.

I have only few concerns.

There is inconsistency in the exchangeable use of the terms “CJD subtypes” and “CJD prions”. 

We apologize for the confusion. In this manuscript “sCJD prions” is used more generally to include all strains and subtypes, whereas “sCJD subtypes” is used when there is a need specify the specific molecular subtypes being referred to. 

MM1 and MV1 are a unique histo-molecular entity (= MM1/MV1 subtype), and transmission studies confirmed their relationship with the same (M1) prion strain. MV2 and VV2 are different CJD subtypes but behave as a unique prion strain (V2) (see https://doi.org/10.1073/pnas.1004688107). Therefore, in the present manuscript the authors evaluated three CJD subtypes (MM1/MV1, MV2 and VV2) and two prion strains (M1 and V2).

We thank the reviewer for making this distinction. We have made note of this in the manuscript.

In the Method section, it is unclear:

1-how many samples/cases did the authors used for each experiment set?

Altogether, 4 human brains, 2 first passage mice, 2 second passage mice, and 1 infected cerebral organoid (as well as negative controls for each) were used. Eight replicates were run for each experimental condition. This has now been clarified in the text.

2-What brain region(s) did the authors sampled for brain homogenates (i.e., CJD subtypes often show a different PrPd distribution).

All human brain samples were derived from the frontal cortex. Mouse tissues were homogenized from one entire hemisphere of the brain. 

3-Did the authors analyzed the PrPd concentration in brain homogenates before beach treatment?

Each specimen that was tested was also titered by end-point dilution analysis. The 50% seeding doses (logSD50/mg) are reported in tables 1 and 2.

Please, check Jacob -> Jakob.

Our apologies for this autocorrect oversite. This has been fixed.

Reviewer #2: In this manuscript, Groveman and colleagues describe the results of an interesting study on the sensitivity of sporadic Creutzfeldt-Jakob disease (sCJD) human prions to bleach inactivation, one of the few effective chemicals for prion decontamination recommended by national and international guidelines.

Authors treated with sodium hypochlorite (9 treatment conditions, 3 times x 3 concentrations) a panel of tissue suspensions derived from sCJD patients and from sCJD-infected humanized mice and organoids. To quantify residual prion infectivity, they measured the amyloid seeding capacity of each sample by the Real-time quaking induced conversion assay (RT-QuIC).

Authors convincingly show that the seeding properties of human and derived prion strains may be fully cleared after the application of bleach, even at conditions that are less stringent than those reported in official guidelines (20.000 ppm per 1 hour; e.g. https://iris.who.int/bitstream/handle/10665/66707/WHO_CDS_CSR_APH_2000.3.pdf?sequence=1&isAllowed=y; Meechan PJ, Potts J. Biosafety in microbiological and biomedical laboratories. 6th ed. Washington, DC: Department of Health and Human Services; 2020). These results remark the usefulness of bleach to improve safety in laboratory, hospital, or industrial settings that have to deal with prion infectivity and provide a novel set of information that, if confirmed and enlarged by further studies, may help, in the future, to develop dedicated decontamination protocols.

The quality and the appropriatedness of the manuscript are beyound any doubt, yet it has to be underlined that its immediate practical consequences, in a delicate field like that of human prion diseases, are limited by the absence of experimental data that relates the RT-QuIC seeding capacity and the “true” infectious titre of sCJD prions for humans.

Also, as bleach treatment may alter RT-QuIC prion seeding capacity in a way that is not directly proportional to its effect on prion infectivity, it would have been useful to compare residual seeding capacity and prion infectivity in at least two dilutions of the same samples after treatment with bleach, following, for example, the approach already used by these authors in previous publications (Hughson AG, Race B, Kraus A, Sangare LR, Robins L, Groveman BR, et al. Inactivation of Prions and Amyloid Seeds with Hypochlorous Acid. PLoS Pathog. 2016;12(9):e1005914).

Some comments on this topic are welcome.

While the experimental data relating seeding activity to infectivity in CJD prions may be limited, there is an abundance of evidence in other prion models that demonstrate a strong and direct correlation between seeding activity and infectivity. In all cases, seeding assays provide equal or greater sensitivity in detection of prions than traditional bioassays with level of seeding activity in strong agreement with infectivity (ie: higher seeding titers correlating with higher LD50s and shorter incubation periods). Additionally, several studies have reported that seeding activity is more reliable and more closely correlated to infectivity than western blotting for PK-resistant PrP. Although we can’t rule out that bleach treatment of CJD prions inhibits seeding activity while leaving infectivity intact, bleach treatment of other prions has been shown to eliminate both seeding activity and infectivity, including in the mentioned Hughson et al. paper. Given the length of time animal bioassays require with untreated 1% CJD brain homogenate (180-450dpi depending on the strain) and the historical data for bleach treatment of other prion strains we felt a parallel bioassay would not strongly enhance our findings in this instance as the use of seeding assays to address question about contamination and decontamination has become fairly widespread and generally well accepted. We thank the reviewer for raising this concern and have added a paragraph to the discussion to address it.

Another issue is linked to the experimental design. In spite of the high sensitivity of RT-QuIC, the experimental setting only allowed to demonstrate a maximum removal of 4 logs of seeding activity for human brain prions and 4.8 logs for mouse-passaged human prions.

We agree that this technical constraint is a drawback to the study. However, we do believe these values to be the minimum possible reduction. Particularly in the cases where no positive wells were detected, our experience tells us that it is unlikely that 100% of the wells would be positive at a ten-fold, or even 100-fold lesser dilution. We agree that these theoretical maximum values should be, and have been, removed from the tables in figures 1 and 2, however we believe that it is important to discuss these reductions as a range rather than just the minimum value. 

All that said, I recommend the publication of this paper with two main suggestions to authors: 

the first, to remove the theoretical values (in parentheses) from tables in figures 1 and 2

This has been done, please see above comment.

 and, the second, to correctly report Alfons Jakob (not Jacob) surname. 

Our apologies again for this autocorrect oversite. This has been corrected.

Only one minor note: in line 151, please specificy the number of supplemental figure.

This has been fixed.

The following references have been added to the manuscript:

Bishop MT, Will RG, Manson JC. Defining sporadic Creutzfeldt-Jakob disease strains and their transmission properties. Proc Natl Acad Sci U S A. 2010;107(26):12005-10. Epub 20100614. doi: 10.1073/pnas.1004688107. PubMed PMID: 20547859; PubMed Central PMCID: PMCPMC2900653.

Vascellari S, Orru CD, Hughson AG, King D, Barron R, Wilham JM, et al. Prion seeding activities of mouse scrapie strains with divergent PrPSc protease sensitivities and amyloid plaque content using RT-QuIC and eQuIC. PLoS One. 2012;7(11):e48969. Epub 2012/11/10. doi: 10.1371/journal.pone.0048969. PubMed PMID: 23139828; PubMed Central PMCID: PMCPMC3489776.

McNulty E, Nalls AV, Mellentine S, Hughes E, Pulscher L, Hoover EA, Mathiason CK. Comparison of conventional, amplification and bio-assay detection methods for a chronic wasting disease inoculum pool. PLoS One. 2019;14(5):e0216621. Epub 20190509. doi: 10.1371/journal.pone.0216621. PubMed PMID: 31071138; PubMed Central PMCID: PMCPMC6508678.

Takatsuki H, Fuse T, Nakagaki T, Mori T, Mihara B, Takao M, et al. Prion-Seeding Activity Is widely Distributed in Tissues of Sporadic Creutzfeldt-Jakob Disease Patients. EBioMedicine. 2016;12:150-5. Epub 20160824. doi: 10.1016/j.ebiom.2016.08.033. PubMed PMID: 27612591; PubMed Central PMCID: PMCPMC5078574.

Simmons SM, Payne VL, Hrdlicka JG, Taylor J, Larsen PA, Wolf TM, et al. Rapid and sensitive determination of residual prion infectivity from prion-decontaminated surfaces. mSphere. 2024:e0050424. Epub 20240827. doi: 10.1128/msphere.00504-24. PubMed PMID: 39189773.

---

## [Editor Report · Decision Letter 1]

15 Oct 2024

Sodium hypochlorite inactivation of human CJD prions

PONE-D-24-29744R1

Dear Dr. Race,

We’re pleased to inform you that your manuscript has been judged scientifically suitable for publication and will be formally accepted for publication once it meets all outstanding technical requirements.

Kind regards,

Gianluigi Zanusso

Academic Editor

PLOS ONE

---

## [Editor Report · Acceptance letter]

22 Oct 2024

PONE-D-24-29744R1 

PLOS ONE

Dear Dr. Race, 

I'm pleased to inform you that your manuscript has been deemed suitable for publication in PLOS ONE. Congratulations! Your manuscript is now being handed over to our production team.

Kind regards, 

on behalf of

Dr. Gianluigi Zanusso 

Academic Editor

PLOS ONE